# *Talaromyces*–Insect Relationships

**DOI:** 10.3390/microorganisms10010045

**Published:** 2021-12-26

**Authors:** Rosario Nicoletti, Andrea Becchimanzi

**Affiliations:** 1Research Centre for Olive, Fruit and Citrus Crops, Council for Agricultural Research and Economics, 81100 Caserta, Italy; 2Department of Agricultural Sciences, University of Naples Federico II, 80055 Portici, Italy; andrea.becchimanzi@unina.it

**Keywords:** crop protection, ecological relationships, entomopathogens, multipurpose biocontrol agents, *Penicillium* subgenus *Biverticillium*, symbiosis

## Abstract

Facing the urgent need to reduce the input of agrochemicals, in recent years, the ecological relationships between plants and their associated microorganisms have been increasingly considered as an essential tool for improving crop production. New findings and data have been accumulated showing that the application of fungi can go beyond the specific role that has been traditionally assigned to the species, employed in integrated pest management as entomopathogens or mycoparasites, and that strains combining both aptitudes can be identified and possibly used as multipurpose biocontrol agents. Mainly considered for their antagonistic relationships with plant pathogenic fungi, species in the genus *Talaromyces* have been more and more widely reported as insect associates in investigations carried out in various agricultural and non-agricultural contexts. Out of a total of over 170 species currently accepted in this genus, so far, 27 have been found to have an association with insects from 9 orders, with an evident increasing trend. The nature of their mutualistic and antagonistic relationships with insects, and their ability to synthesize bioactive compounds possibly involved in the expression of the latter kind of interactions, are analyzed in this paper with reference to the ecological impact and applicative perspectives in crop protection.

## 1. Introduction

Traditionally, the role of fungi in the contrasting biological adversities of plants has been considered with reference to their assignment to functional categories, such as ‘mycoparasites’, ‘entomopathogens’, ‘nematophagous fungi’, etc. More recently, the diffusion of the holistic approach considering the ecological interrelations among plant-associated organisms has impaired such a rigid distinction. Indeed, the recurring evidence that many fungi can play several mutualistic roles stimulates a reappraisal of their effects on plant health, as well as of the way they can be employed in the integrated management of pests and pathogens of crops [1]. A number of widespread entomopathogens, such as *Beauveria bassiana* and *Lecanicillium/Akanthomyces* spp., have also qualified as being able to perform antagonism against many plant pathogens, so that they can be employed as dual or multipurpose biocontrol agents [2,3,4]. Likewise, *Trichoderma* spp., until recently essentially regarded as fungal antagonists and mycoparasites, have also started to be considered for the role they are able to play against several pests [5,6].

Current evidence concerning the capacity of *Talaromyces*, another Ascomycetes genus including several species reported as antagonists of plant pathogens [7,8,9], to directly interact with insects is examined in this paper in view of a more comprehensive consideration of their role in crop protection and possible applications in the biocontrol of these key pests.

## 2. Occurrence of *Talaromyces* spp. as Insect Associates

Limited insights have been published on the association of *Talaromyces* spp. (Eurotiales, Trichocomaceae) with insects, depending on their traditional categorization in the guild of fungal antagonists. As well as the increasing awareness of their ability to play multiple ecological roles, the recent accumulation of data on these Ascomycetes is due to the introduction of molecular tools in fungal taxonomy. In fact, until the enunciation of the principle ‘one fungus, one name’ about 10 years ago, the genus name *Talaromyces* was basically used for symmetrically biverticillate *Penicillia* producing a perfect stage in axenic culture. Conversely, species for which no teleomorph could be observed were classified in the polyphyletic genus *Penicillium*. Phylogenetic analyses based on molecular markers pointed out this inconsistency, so that species in the *Penicillium* subgenus *Biverticillium* were officially separated and formally assigned to *Talaromyces* [10]. At the same time, the spread of DNA sequencing in the identification of fungi collected from any ecological context has enabled their correct (or less approximate) classification, and provided a remarkable boost to investigations aiming at a general description of the mycobiomes of plants and other organisms.

The references provided in Table 1, which do not include cases of mycophagy, confirm the increasing trend in reports of *Talaromyces* spp. in associations with insects. In fact, the first documented finding dates back to 1990, when isolates of *Talaromyces funiculosus* (at that time known as *Penicillium funiculosum*) were recovered from larvae of *Aedes cantans* and *Aedes communis* (Diptera, Culicidae) collected in the Kiev region, Ukraine [11]. Three more reports were published in the 1990s, concerning the finding of *Talaromyces diversus* (identified as *Penicillium diversum*) and *Talaromyces trachyspermus* from meconia in nests of the paper wasp *Polistes hebraeus* (= *P*. *olivaceus*) (Hymenoptera, Vespidae) in La Reunion island [12], *Talaromyces purpureogenus* (identified as *Penicillium purpurogenum*) on several species of mosquitoes in Brazil [13] and *Talaromyces duclauxii* (as *Penicillium duclauxii*) from dead larvae and adults of the cave cricket *Troglophilus neglectus* (Orthoptera, Rhaphidophoridae) in Slovenia [14]. All other findings have been reported in papers published after 2004; after having been quite infrequent until 2015, the number of articles has more than doubled in the last five years.

Out of a total of over 170 species accepted in the last genus revision [56], just 27 have been reported in associations with insects so far. However, some of these findings refer to new or infrequent species, supporting the expectation that exploring this symbiotic context can be fruitful in terms of unveiling hidden biodiversity. In fact, *Talaromyces cecidicola* was originally described as *Penicillium cecidicola* based on isolates from galls formed by unidentified cynipid wasps on scrub oaks (*Quercus pacifica*) in the western United States [18]. This species is sister to *Talaromyces dendriticus* (originally described as *Penicillium dendriticum*), which was also isolated from galls produced by an unidentified insect on *Eucalyptus* leaves in Australia [18]. Other gallicolous species, *Talaromyces erythromellis* and *Talaromyces pseudostromaticus* (again both originally described as *Penicillium* species), were identified from galls of another cynipid wasp (*Diplolepis rosae*) on *Rosa sitchensis* in Canada [18].

Besides gallicolous cynipids, interactions with Hymenoptera have resulted from investigations concerning nests built by some eusocial species of these insects. These include the above-mentioned finding of *T*. *diversus* and *T*. *trachyspermus* from nests of *P*. *hebraeus* [12], *Talaromyces variabilis* (currently synonymized with *Talaromyces wortmannii*) from nests of another paper wasp (*Ropalidia marginata*) [54] and three new species (*Talaromyces brasiliensis*, *Talaromyces pigmentosus* and *Talaromyces mycothecae*), which have been described based on isolates recovered from nests of the Brazilian stingless bee *Melipona scutellaris* (Hymenoptera, Apidae), along with the known species *Talaromyces scorteus* and *T*. *wortmannii* [17]. Other records come from the nests of several species of leaf-cutting ants in Texas (USA) [33], Brazil [50] and Argentina [37]. However, at least in the case of the anthill gardens, the occurrence of three common species (*Talaromyces rugulosus*, *Talaromyces verruculosus* and *T*. *purpureogenus*) does not seem to be related to any specific function, rather depending on saprophytism or mycoparasitism.

Uncircumstantial associations with other well-known eusocial insects, the termites (Blattodea, Termitidae and Rhinotermitidae), have been documented for a few isolates: *Talaromyces flavus* from the coarse debris of wood infested by *Reticulitermes* sp. in Mississippi (USA) [24], *Talaromyces* spp. from the nests of *Coptotermes formosanus* under experimental rearing conditions in Florida (USA) [40] and combs of the fungus-growing species *Macrotermes carbonarius* in Vietnam [39], while *Talaromyces stollii* was reported from the combs of *Macrotermes barneyi* in China [47].

All other findings concern living or dead insects belonging to Coleoptera, Diptera, Hemiptera, Hymenoptera, Lepidoptera, Orthoptera, Thysanoptera and Trichoptera. Each of the last three orders has just a single occasional citation concerning the above-mentioned strains of *T*. *duclauxii* from *T*. *neglectus* [14], a strain of *T*. *verruculosus* from an unidentified thrips in Thailand [49] and a strain of *T*. *purpureogenus* from the gut of a larva of an aquatic shredder (*Triplectides* sp.) (Trichoptera, Leptoceridae) in Amazonia (Brazil). The latter strain displayed cellulolytic properties, which can be inferred to possibly contribute to the digestion of plant matter [36]. Conversely, more systematic and abundant occurrences are those of *Talaromyces* spp. in the abdomen fat body and gut of the brown planthopper (*Nilaparvata lugens*) (Hemiptera, Delphacidae) in China [45,46], which deserves further investigation in view of the assessment of eventual functional relationships. Moreover, the occurrence of *Talaromyces* in low relative abundance resulted in the analysis of the endomycobiome of another planthopper species, *Delphacodes kuscheli* [57].

A functional role based on cellulolytic properties could also be implicated in the association of *Talaromyces* spp. with xylicolous Coleoptera. This is the case of *Talaromyces pulveris*, which has been very recently described based on an isolate from bore dust of the deathwatch beetle (*Xestobium rufovillosum*, Anobiidae) in France [32], while other findings concern the gut of the ribbed pine borer (*Rhagium inquisitor*, Cerambycidae) collected in Tatarstan (Russia) [43] and of the ambrosia/bark beetles (Curculionidae, Platypodinae and Scolytinae). In particular, many cases are known for species in the latter group, starting from *Polygraphus poligraphus* and *Ips* spp. infesting coniferous forests in the Czech Republic where two species *(T*. *rugulosus* and *T*. *variabilis*) were represented in the core mycobiome, as well as *Talaromyces minioluteus*, which was only isolated from *Ips sexdentatus* [29]. In fact, other strains of *Talaromyces atroroseus* and *Talaromyces pinophilus*, respectively recovered from living and dead adults of the pistachio bark beetle (*Chaetoptelius vestitus*) in Tunisia, have been characterized with reference to cellulase and protease production [16]. More findings concern a strain of unidentified species from living adults of the pinhole borer (*Platypus cylindrus*) infesting oaks at the Astroni Nature Reserve in southern Italy (Nicoletti, unpublished), multiple isolates of *Talaromyces radicus*, *T*. *variabilis* and *T*. *purpureogenus* from *Dendroctonus* spp. associated with coniferous plants in France [34], *T*. *trachyspermus* from adults of *Ips typographus* trapped in Norway spruce stands in northwestern Italy [48], *T*. *minioluteus* and *T*. *purpureogenus* from adults of *Ips acuminatus* collected in the Ukraine [28], and *T*. *verruculosus* from the mycangia of adults of *Xylosandrus crassiusculus* from specimens collected in Florida (USA) and China [51]. The epibiotic association with three *Tomicus* species (*T*. *brevipilosus*, *T*. *minor* and *T*. *yunnanensis*) of multiple *Talaromyces* spp. anonymously identified as ‘OTUs’ has been reported from the Yunnan province of the latter country [44]. Moreover, in a similar study carried out in Mexico, a high prevalence of *T*. *purpureogenus* was observed in association with *Xyleborus affinis* and *Xyleborus bispinatus* [35]. Finally, the occurrence of *T*. *rugulosus* was detected in gardens of *Xyleborinus saxesenii* under laboratory rearing conditions [38].

Isolations from the gut were also obtained in the case of haematophagous species such as the kissing bug *Triatoma infestans* (Hemiptera, Reduviidae) from Argentina [27] and the mosquito *Aedes aegypti* from Puerto Rico [41], both deserving further investigation. In fact, the discovery of three species (*Talaromyces ruber*, *Talaromyces* aff. *helicus* and *T*. *purpureogenus*) on *T*. *infestans* is indicative of a possible natural spread of *Talaromyces* in association with these noxious bugs. Moreover, a strain of unidentified species from *A*. *aegypti* was found to facilitate the infection of mosquitoes with the dengue virus (DENV), thereby possibly enhancing virus transmission. This modulation is related to the down-regulation of digestive enzyme genes and trypsin activity upon exposure to factors secreted by the fungus in the insect’s gut. The experimental finding that the fungus can be acquired by mosquitoes through sugar feeding, and that it is able to successfully colonize the midgut for a period of at least 25 days, is indicative of a functional relationship that requires better elucidation [41]. Uncircumstantial relationships with a blood sucking insect have also been considered with regard to isolates of *T*. *variabilis* from the larvae of *Simulium goeldii* (Diptera, Simuliidae) in Amazonia (Brazil) [53].

Other findings are seemingly occasional and do not represent indications of possible functional relevance. This is the case of two Egyptian reports concerning *T*. *funiculosus* from dead larvae and pupae of the vegetable leafminer (*Liriomyza sativae*) (Diptera, Agromyzidae) [26] and *T*. *pinophilus* from an unspecified stage of the red palm weevil (*Rhynchophorus ferrugineus*) (Coleoptera, Curculionidae) [30], as well as of *Talaromyces* sp. from an unidentified mud dauber wasp in Australia [42], *Talaromyces versatilis* from dead individuals of the giant honey bee (*Apis dorsata*) (Hymenoptera, Apidae) in Sri Lanka [52] and *T*. *variabilis* from adults of *Diabrotica* sp. (Coleoptera, Chrysomelidae) and *Neomyopites* sp. (Diptera, Tephritidae) collected on the subshrub *Espeletia pycnophylla* in Colombia [55]. In the course of an investigation carried out in Lebanon, the species *Talaromyces amestolkiae* exhibited the capacity to adapt to insect hosts from different orders, with several strains recovered from corpses of *Capnodis tenebrionis* (Coleoptera, Buprestidae), *Culex* sp. (Diptera, Culicidae) and unidentified species of Pyrrhocoridae (Hemiptera) and Pyralidae (Lepidoptera) [15]. Concerning the last order, two strains of *T*. *flavus* and several strains of *T*. *pinophilus* were recovered from larvae and pupae of the antophagous and carpophagous generations of the olive moth (*P*. *oleae*) (Plutellidae) in Portugal [23]. Moreover, strains of *T*. *trachyspermus* and *T*. *flavus* were recovered in China from larvae of the honeycomb moth (*Galleria mellonella*) (Pyralidae) used as bait [21,22], and the latter species was found in frass collected in tunnels of the maize stalk borer (*Busseola fusca*) (Noctuidae) in South Africa [25]. Finally, *T*. *flavus* was reported from mines of the long-legged flies *Thrypticus truncatus* and *Thrypticus sagittatus* (Diptera, Dolichopodidae) on water hyacinth (*E*. *crassipes*) in Argentina [20].

## 3. Experimental Evidence of Anti-Insectan Effects

It is quite obvious that the isolation of fungi from insects and/or their nests merely represents an indication of ecological interactions that do not necessarily have adaptive or functional implications. However, the intent to exploit anti-insectan properties in biocontrol has stimulated investigations aimed at assessing if the observed associations may eventually result in detrimental effects on several insect pests.

Experimental assays were carried out with a strain of *T*. *flavus* isolated from larval breeding sites of the mosquito *Anopheles albimanus* (Diptera, Culicidae) in the coastal plain of Chiapas (Mexico), which induced low mortality (16.6%) on nymphs of the kissing bug (*Triatoma dimidiata*). Nymphs died after a long period following inoculation (21 days), and the fungus exhibited no sporulation; however, when administered to adults, mortality rose to 75%, with a lower interval (16 days). This difference possibly derived from the loss of conidia, which may have occurred during molting after the topical inoculation of nymphs [58].

The degree of susceptibility to the anti-insectan effects of *Talaromyces* may depend on the insect species and may be affected by the experimental conditions. In preliminary tests, high concentrations of conidia of strains of *T*. *funiculosus* killed second instar larvae of the mosquitoes *Culex pipiens* ‘form *molestus*’ and *A*. *aegypti* [11]. Conidial suspension of another strain of this species affected vitality and oviposition in assays carried out against *L*. *sativae* on tomato [26]. Strains of *T*. *amestolkiae* recovered from the corpses of several insects were found to only slightly affect the survival of adults of the vinegar fly (*Drosophila melanogaster*) (Diptera, Drosophilidae) and the tiger mosquito (*Aedes albopictus*) following inoculation under laboratory conditions [15]. A strain of *T*. *verruculosus* isolated from an unidentified thrips caused mortality at a low level (13–23%) in bioassays carried out on thrips (*Ceratothripoides claratris*) (Thysanoptera, Thripidae), mealybugs (*Pseudococcus cryptus*) (Hemiptera, Pseudococcidae) and whiteflies (*Bemisia tabaci*) (Hemiptera, Aleyrodidae) [49]. A low rate of mortality (about 25%) was also caused by another isolate of *T*. *verruculosus* from larva of *Bactrocera oleae* (Diptera, Tephritidae) in assays carried out on larvae of the Mediterranean flour moth (*Ephestia kuehniella*) (Lepidoptera, Pyralidae) [31]. Finally, in laboratory assays carried out on the black bean aphid (*Aphis fabae*) and the Russian wheat aphid (*Diuraphis noxia*), a strain of *T*. *pinophilus* recovered from *R*. *ferrugineus* caused 30% and 50% mortality, respectively [30].

Experimental evidence of anti-insectan effects was also achieved in investigations concerning *Talaromyces* strains obtained from other sources. This is the case of *T*. *minioluteus* found in the laboratory diet of the spotted wing drosophila (*Drosophila suzukii*), which affected flies’ development by extending the pupal stage by 16.22%, shortening adult longevity by 15.52% and reducing survival by 71.67% [59]. Moreover, a few strains recovered from soil in several locations in Indonesia yielded quite positive results. In particular, strains of *Talaromyces sayulitensis* from the rhizosphere of pineapple, corn and pepper were found to be able to infect and cause the mortality (16.67–46.67%) of cocoa bugs (*Helopeltis* sp.) (Hemiptera, Miridae) [60,61]. Moreover, a strain of *T*. *diversus*, isolated from soil cropped to mustard using mealworm larvae (*Tenebrio molitor*) (Coleoptera, Tenebrionidae) as bait, was found to possess insecticidal activity as assayed by both conidial suspension and culture filtrate against the cotton leafworm (*Spodoptera litura*) (Lepidoptera, Noctuidae) [19], while the use of a strain of *T*. *pinophilus* from a rubber tree plantation was proposed in the formulation of a bioinsecticide for the control of the coffee borer beetle (*Hypothenemus hampei*) (Coleoptera, Curculionidae, Scolytinae) [62]. Finally, a strain of *T*. *verruculosus* recovered from soil in West Bengal was reported for its notable entomopathogenic effects in laboratory assays carried out against the cotton aphid (*Aphis gossypii*) (Heteroptera, Aphididae). It must be noted that, probably due to species name similarity, this strain was incorrectly reported as *Penicillium verrucosum*, despite evidence concerning its morphology and ITS sequencing provided in the original report being clearly indicative of *T*. *verruculosus* [63]. In this respect, it is recommended that the authors make the necessary amendment to the data deposited in GenBank in order to avoid that further misleading identifications may occur in the future.

## 4. The Role of Secondary Metabolites

Despite the recognized relevance of bioactive secondary metabolites in shaping the relationships between fungi and arthropods [64], none of the above-mentioned studies addressed whether the observed anti-insectan properties are due to the release of these products by the insect-associated fungi. Indeed, *Talaromyces* species are renowned as producers of a wide array of bioactive compounds [65,66,67], and some clues that insect-associated strains may also represent a source of bioactive products have also arisen. This is shown in the case of the hymenopteran-derived strains of *T*. *versatilis* and *Talaromyces* sp. that display antibacterial properties in their culture extracts or purified compounds [52,68], and the three new species described from isolations from the nests of *M*. *scutellaris*. Among the latter, *T*. *mycothecae* produced the isocoumarin dimer antibiotic duclauxin (Figure 1) and some derivatives of the same, while *T*. *brasiliensis* and *T*. *pigmentosus* were found to produce several unknown secondary metabolites [17].

Conversely, several studies involving *Talaromyces* strains obtained from other sources have provided more circumstantial evidence that these fungi may produce anti-insectan compounds. In fact, dichloromethane and methanol extracts from cultures of a strain of *T*. *funiculosus* displayed various activities in assays carried out on insects; this includes aphicidal activity against the green peach aphid (*Myzus persicae*) (Hemiptera, Aphididae), ovicidal activity against the Colorado potato beetle (*Leptinotarsa decemlineata*) (Coleoptera, Chrysomelidae), adulticide activity against the large milkweed bug (*Oncopeltus fasciatus*) (Hemiptera, Lygaeidae) and juvenile hormone mimetic activity on the German cockroach *Blattella germanica* (Blattodea, Ectobiidae) [69]. Moreover, dichloromethane extracts obtained from the cultures of strains of *T*. *funiculosus*, *T*. *purpureogenus* and *T*. *rugulosus* from cereal grains displayed various degrees of toxicity in assays carried out on nymphs of *O*. *fasciatus* [70].

Similar evidence has also been obtained for some purified compounds (Figure 1). For instance, phlegmacin B_1_ from a soil strain of *Talaromyces* sp., a dimeric pre-anthraquinone possessing inhibitory properties towards chitinases of the Asian corn borer (*Ostrinia furnacalis*) (Lepidoptera, Crambidae), may impair larval development during molting [71]. Moreover, 3-O-methylfunicone produced by *T*. *pinophilus*, a benzo-γ-pyrone known for its notable antifungal and antitumor properties [72,73], displayed aphicidal effects as assayed on the pea aphid (*Acyrthosiphon pisum*) (Hemiptera, Aphididae) [74].

Anti-insectan properties have been investigated in more detail in the case of chrodrimanins, meroterpenoids produced by several *Talaromyces* spp. [75,76,77,78,79]. Chrodrimanins B (Figure 1), D, E and F exhibited insecticidal activity when added to the diet of third instar larvae of silkworm (*Bombyx* *mori*) (Lepidoptera, Bombycidae), with LD_50_ values of 10, 20, 10 and 50 µg/g, respectively [77,80]. Based on the observation that the treated larvae underwent paralysis, the bioactivity of chrodrimanin B was investigated using patch-clamp electrophysiology on ligand-gated ion channels of larval neurons. The compound was found to have no effect on membrane currents when tested at 1 μM. However, when delivered at the same concentration for 1 min prior to co-application with *γ*-aminobutyric acid (GABA), it completely blocked the GABA-induced current, also displaying minor actions on acetylcholine- and l-glutamate-induced currents. Moreover, chrodrimanins A, B and D were also tested on a wild-type isoform of the GABA receptor of silkworms and were found to be able to attenuate the peak current amplitude of the GABA response, with an IC_50_ of 1.66 nM. The order of the blocking potency of chrodrimanins (B > D > A) was in accordance with their reported insecticidal effects. Chrodrimanin B appears to be a selective blocker of insect GABA receptors since its activity on the human GABA receptor was approximately 10^3^-fold lower [81].

Pyripyropene A (Figure 1) is another meroterpenoid reported as a secondary metabolite in *Talaromyces* [82]. This product displayed high activity against *M*. *persicae*, not only in laboratory assays at a concentration of 1.25 ppm but also in applications on cabbage plants through both foliar sprays and soil drenching. Structure-related bioactivity studies showed that the pyridine nucleus of this compound is an important pharmacophore since its replacement with a phenyl ring in the analogue phenylpyropene A caused the loss of insecticidal activity. Moreover, the effectiveness of derivatives lacking the acetyl ester groups was remarkably lower, suggesting the possibility of improving bioactivity by modifying these parts of the molecule [83]. The compound confirmed good activity in further assays carried out on *A*. *gossypii*, the mealybug *Pseudococcus comstocki* and two whiteflies (*B*. *tabaci* and *Trialeurodes vaporariorum*) (Hemiptera, Aleyrodidae), while it was inactive against the rice leaf bug (*Trigonotylus* *caelestialium*) (Hemiptera, Miridae), the planthoppers *N*. *lugens* and *Nephotettix cincticeps* (Hemiptera, Cicadellidae), the diamondback moth (*Plutella xylostella*) (Lepidoptera, Plutellidae), *S*. *litura* and the western flower thrips (*Frankliniella occidentalis*) (Thysanoptera, Thripidae) [84].

Besides evidence of its direct insecticidal effects, pyripyropene A has been considered as a model for the synthesis of molecules with improved bioactivities [85,86,87]. In this respect, it has also been characterized for acetyl-CoA acyl transferase inhibitory effects, which are indicative of its potential insecticidal properties [88]; however, the above-mentioned study by Horikoshi et al. [83] pointed out that this mechanism is not able to explain the observed bioactivity. On the other hand, the culture extracts of many fungi have been found to possess acetylcholinesterase (AchE) inhibitory properties, which is considered a primary target for insecticides [89]. The production of AchE inhibitors has also been reported for some *Talaromyces* strains [23,90], and some products have been purified and characterized for this bioactivity, such as talaromycesone A and talaroxanthenone [91].

Initially reported as a coloring matter produced by *Penicillium rugulosum* (= *T*. *rugulosus*) [92], rugulosin (Figure 1) is another product that has been thoroughly investigated for its insecticidal properties. In fact, this bis-anthraquinone was first characterized for toxicity against *D*. *melanogaster* (ED_50_ 27.6 µg/mL) [93]. Later on, toxic effects were described on Sf9, a cell line derived from ovarian cells of the fall armyworm (*Spodoptera* frugiperda) (ID_50_ 1.2 µg/mL) [94] and in direct feeding assays on larvae of the spruce budworm (*Choristoneura fumiferana*) (Lepidoptera, Tortricidae) [95,96]. Moreover, toxic effects resulted when larvae of the latter insect fed on the needles of white spruce (*Picea glauca*) colonized by a rugulosin-producing endophytic fungus, which contained the toxin at a concentration that was effective at retarding the larval growth in vitro [97,98]. In line with the presumptive role of anthraquinones in defensive mutualism established between plants and endophytic fungi [99], and with reference to the known endophytic occurrence of *T*. *rugulosus* and other *Talaromyces* spp. producing these compounds [100,101,102], this mechanism could affect insect herbivory and protect plants where these fungi are able to develop endophytically. It is interesting to note that in the study by Miller et al. [97], a similar effect was observed after the inoculation of spruce needles with an unidentified endophytic strain producing vermiculine (Figure 1), a macrodiolide antibiotic that so far has only been reported from *T*. *flavus* [103].

## 5. Conclusions

The analysis of the available literature has disclosed that many *Talaromyces* spp. are able to establish ecological relationships with insects. At least 8 out of 27 species have been first or exclusively recovered from entomological specimens, indicating that insects represent a source of unknown biodiversity with reference to this fungal genus. Indeed, the recovery of *Talaromyces* strains even from freshwater and troglophile insects represents evidence that more cases of interesting associations could be disclosed as a result of investigations carried out in diverse ecological contexts.

However, in most instances, the reported interactions must be recorded as neutral, in the absence of circumstantial evidence clearly referable to either mutualistic or antagonistic effects. Apart from citations merely concerning occurrence, to be considered as occasional unless corroborated by additional findings in the future, the symbiotic interactions could be inferred as being mutualistic in the case of gut-associated strains that might contribute to the digestion of specific feed. Conversely, in addition to the cases where detrimental effects on insect viability have been experimentally demonstrated, indirect antagonistic behavior can be conjectured in the case of species/strains found in the gardens of leaf-cutting ants and in combs of *Macrotermes* spp. with reference to mycoparasitism possibly exerted against the cropped fungi.

Antagonistic interactions of fungi are often mediated by the production of bioactive metabolites, which could either affect the insect’s development when direct contact is established or be responsible for toxic or phagodeterrent effects when these products are released in plants by endophytic strains. Indeed, this aspect may have a substantial impact in crop protection, with reference to the possibility to exploit the resident mycoflora and/or artificially-administered strains holding this property. Species such as *T*. *flavus*, *T*. *pinophilus* and *T*. *purpureogenus* are already known to be producers of bioactive compounds, playing a fundamental role in antagonism exerted against plant pathogens. Considering that these species are also increasingly reported as plant growth promoters [104], more accurate investigations concerning their anti-insectan aptitude might disclose additional positive effects on plant health and integrate the profile of multipurpose strains to be employed in crop management.

## Figures and Tables

**Figure 1 microorganisms-10-00045-f001:**
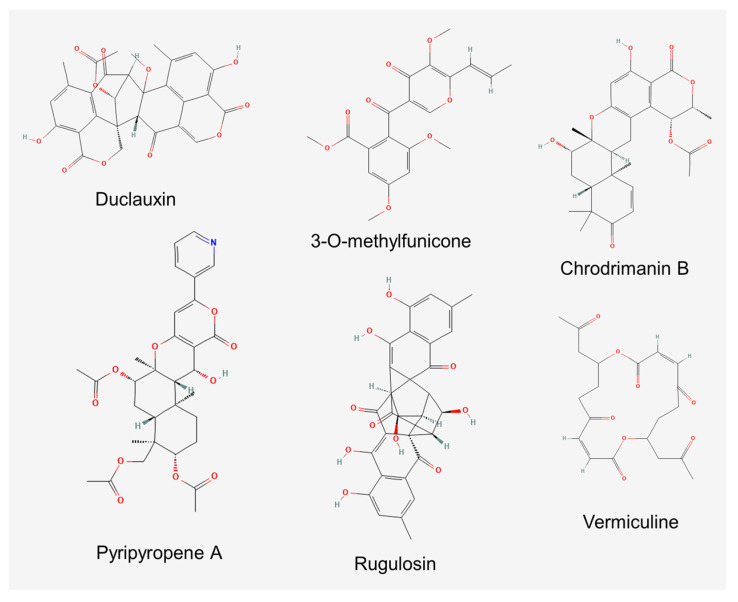
Molecular structure of anti-insectan products of *Talaromyces* spp.

**Table 1 microorganisms-10-00045-t001:** Occurrence of *Talaromyces* species found in association with insects.

Species	Insect Source *	Location	Year	Reference
*T*. *amestolkiae*	corpses of *Capnodis tenebrionis*, *Culex* sp., unidentified Pyralidae, unidentified Pyrrhocoridae	Lebanon	2016	[15]
*T*. *atroroseus*	living adults of *Chaetoptelius vestitus*	Tunisia	2020	[16]
*T*. *brasiliensis*	nests of *Melipona scutellaris*	Brazil	2018	[17]
*T*. *cecidicola*	galls of unidentified cynipid on *Quercus pacifica*	Western USA	2004	[18]
*T*. *dendriticus*	galls of unidentified insect on *Eucalyptus* leaves	Australia	2004	[18]
*T*. *diversus*	meconia of *Polistes hebraeus*	La Reunion	1995	[12]
	larva of *Tenebrio molitor* (bait)	Sumatra	2020	[19]
*T*. *duclauxii*	dead larvae and adults of *Troglophilus neglectus*	Slovenia	1998	[14]
*T*. *erythromellis*	galls of *Diplolepis rosae* on *Rosa sitchensis*	Canada	2004	[18]
*T*. *flavus*	mines of *Thrypticus truncatus*, *Thrypticus sagittatus* on *Eichhornia crassipes*	Argentina	2007	[20]
	larvae of *Galleria mellonella* (bait)	China	2008	[21,22]
	larvae of *Prays oleae*	Portugal	2012	[23]
	coarse wood debris with termites (*Reticulitermes* sp.)	Mississippi	2012	[24]
	frass of *Busseola fusca*	South Africa	2020	[25]
*T*. *funiculosus*	larvae of *Aedes cantans* and *Aedes communis*	Ukraine	1990	[11]
	dead larvae and pupae of *Liriomyza sativae*	Egypt	2006	[26]
*T*. aff. *helicus*	gut of *Triatoma infestans*	Argentina	2007	[27]
*T*. *minioluteus*	adults of *Ips acuminatus*	Ukraine	2017	[28]
	gut of *Ips sexdentatus*	Czechia	2020	[29]
*T*. *mycothecae*	nests of *M. scutellaris*	Brazil	2018	[17]
*T*. *pigmentosus*	nests of *M. scutellaris*	Brazil	2018	[17]
*T*. *pinophilus*	larvae of *P*. *oleae*	Portugal	2012	[23]
	*Rhynchophorus ferrugineus*	Egypt	2019	[30]
	larvae of *Bactrocera oleae*, *Euphyllura olivina*	Tunisia	2020	[31]
	dead adults of *C*. *vestitus*	Tunisia	2020	[16]
*T*. *pseudostromaticus*	galls of *D*. *rosae* on *R*. *sitchensis*	Canada	2004	[18]
*T*. *pulveris*	bore dust of *Xestobium rufovillosum*	France	2020	[32]
*T*. *purpureogenus*	mosquitoes (*Aedes* sp., *Anopheles* sp., *Mansonia* sp.)	Brazil	1998	[13]
	gut of *T*. *infestans*	Argentina	2007	[27]
	nest of *Trachymyrmex septentrionalis*	Texas	2011	[33]
	larvae of *Dendroctonus punctatus*	France	2016	[34]
	adults of *I*. *acuminatus*	Ukraine	2017	[28]
	adults of *Xyleborus affinis*, *Xyleborus bispinatus*	Mexico	2018	[35]
	gut of larva of *Triplectides* sp.	Brazil	2020	[36]
*T*. *radicus*	larvae of *D*. *punctatus*	France	2016	[34]
*T*. *ruber*	gut of *T*. *infestans*	Argentina	2007	[27]
*T*. *rugulosus*	nests of *Acromyrmex* spp.	Argentina	2017	[37]
	gut of *Ips duplicatus*, *Ips typographus*, *I*. *acuminatus*, *I*. *sexdentatus*, *Polygraphus poligraphus*	Czechia	2020	[29]
	gardens of *Xyleborinus saxesenii* in laboratory rearing	Switzerland	2021	[38]
*T*. *scorteus*	nests of *M*. *scutellaris*	Brazil	2018	[17]
*Talaromyces* sp.	fungal comb in nest of *Macrotermes carbonarius*	Vietnam	2019	[39]
	nest of *Coptotermes formosanus*	Florida	2013	[40]
	gut of *Aedes aegypti*	Puerto Rico	2017	[41]
	unidentified mud dauber wasp	Australia	2017	[42]
	gut of larvae of *Rhagium inquisitor*	Russia	2018	[43]
	adults of *Tomicus brevipilosus*, *Tomicus minor*, *Tomicus yunnanensis*	China	2020	[44]
	adult of *Platypus cylindrus*	Italy	2020	This paper
	abdomen fat body of *Nilaparvata lugens*	China	2021	[45]
	gut of *N*. *lugens*	China	2021	[46]
*T*. *stollii*	fungal comb in nest of *Macrotermes barneyi*	China	2021	[47]
*T*. *trachyspermus*	meconia of *P*. *hebraeus*	La Reunion	1995	[12]
	larvae of *G*. *mellonella* (bait)	China	2008	[21]
	adults of *I*. *typographus*,	Italy	2013	[48]
*T*. *verruculosus*	unidentified thrips	Thailand	2007	[49]
	nests of *Atta texana, Cyphomyrmex wheeleri, T. septentrionalis*	Texas	2011	[33]
	nests of *Atta cephalotes*	Brazil	2015	[50]
	mycangia of *Xylosandrus crassiusculus*	China, Florida	2019	[51]
	larvae of *B*. *oleae*, *E*. *olivina*	Tunisia	2020	[31]
*T*. *versatilis*	dead adult of *Apis dorsata*	Sri Lanka	2020	[52]
*T*. *wortmannii**(T*. *variabilis)*	larvae of *Simulium goeldii*	Brazil	2008	[53]
	nest of *Ropalidia marginata*	India	2010	[54]
	larvae and adults of *Dendroctonus micans*, *Dendroctonus valens*	France	2016	[34]
	nests and honey of *M*. *scutellaris*	Brazil	2018	[17]
	gut of *Ips cembrae*, *I*. *acuminatus*, *I*. *duplicatus*, *I*. *sexdentatus*, *I*. *typographus*	Czechia	2020	[29]
	adults of *Diabrotica* sp. and *Neomyopites* sp.	Colombia	2020	[55]

* Colors are indicative of the order to which the species belong, as follows: Blattodea; Coleoptera; Diptera; Hemiptera; Hymenoptera; Lepidoptera; Orthoptera; Thysanoptera; Trichoptera.

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
