# Peer review of "Talaromyces–Insect Relationships"

_microorganisms, 2021, doi:10.3390/microorganisms10010045_

Round 1
Reviewer 1 Report
The manuscript represents an extensive list of records on species currently accepted in the genus Talaromyces identified mostly from insects. The work is based on coprehensive overview of relevant literature, at least from the recent decade when more precise species identification has been possible.
The work would benefit from adding a phylogenetic aspect to the review to address some basic questions, e.g. whether the transition to insect hosts (solely or among others) has likely been a single event in Talaromyces or the family Aspergillaceae or this ability has been acquired in several closely/distantly related groups? Why was Talaromyces picked for the review among the respective family, do members of related genera also include members recorded as entomopathogens?
This could be presented as annotations on a previously published phylogenetic tree or a discussion comparing the cited fungal species and their hosts. That also would allow the authors to comment the degree of host specificity by looking whether some Talaromyces lineages have been reported from particular insect orders.
The current abstract does not summarize the review (except for the last sentence) but rather the introduction. The proportion of these two parts should be opposite.
It might suffice to say somewhere in the beginning that most of the species treated in this study were initially described in Penicillium and then not to report the old synonyms in each case.
While the language is generally good, several sentences are a bit too sophisticated and could be made easier to follow. Such places as well as some errors and further suggestions to improve the text are following.
Replace following words by such that would be more appropriate in the given context:
line 15 'basically'
32 'to contrast'
42-45 'assessments' of what? this sentence is long and hard to understand
45 'removal of the above cultural constraint, the change is consequential to' replace by shorter and more clear phrase
49 'failed to differentiate the teleomorph went on being' -> 'species for which no teleomorph was observed were classified in Penicillium
75 'quite original' and the whole sentence is awckward and should be simplified
97 'seems' > seem
99 add hyphen in between 'well known'
106 'individuals of species' -> insects (Collembola has been included among the hosts described in the review. These, however, are not considered as part of Insecta, in current taxonomy.)
118 how did Talaromyces occurrence result in the given analysis?
122 delete 'the new species'
141 add 'USA* after Florida
148 if this whole paragraph is about isolates from guts then inform the reader about that first with an introductory sentence.
172 delete 's' at the end of investigation
173 'of' -> 'from'
176 say 'Talaromyces species have been isolated from members of Lepidoptera in ...' and divide this extremely long sentence into two or three.
181 delete 'again'
182 delete 'once more'
187 add 'of fungi' after 'isolation from'
199 delete 'Of course,'
242-244 write 'studies address whether the observed ...'
245 'limited evidence has arisen in the case of' is just overcomplicated, say it in a simple way
343 evidence should be used in singular
344 Apart FROM, rubricated -> considered
Author Response
Thank you for your help and useful suggestions for revising our manuscript. Please see the attachment.

Reviewer 2 Report
Dear Authors, your ms it is very good especially at the part 2,3 and 4. The Introduction and the conclusion are not at the level of the good parts. The conclusion must be written again. Please write again the conclusion
Author Response
Thank you for your positive judgment. Please see the attachment.

Round 2
Reviewer 1 Report
The authors have provided satisfying answers to the questions and made the suggested changes in the text. A few issues, partly introduced along with the amendments, should still be fixed:
Lines 20-21 Delete the added part of the sentence ‘which is indicative of possible new findings deriving from the exploration of this particular symbiotic context.’ as this isn’t adding any information.
49 carried out -> published Delete ‘in the past’
51-54 I still don’t get what is meant by ‘depending on their traditional categorization in the guild of fungal antagonists’ and ‘'removal of the above cultural constraint’ Do you mean that Talaromyces/Penicillium species were traditionally considered as saprotrophs and therefore, even if found on insects, these were not classified as potential entomopathogens or insect mutualists? What has changed in this regard? I don’t think this can be called ‘cultural’ but rather an improvement of our knowledge based on increasing sampling and improvement of identification methods. Please change in a way allowing unambiguous understanding.
the molecular tool -> molecular tools
The sentence on lines 129-131 is impossible to understand, especially now after an addition to its end:
Moreover, the occurrence of Talaromyces resulted in the analysis of the endomycobiome of another planthopper species, Delphacodes kuscheli, even if in low relative abundance [57].
Who was observed in relative abundance and how did the observation of the fungus result in anayzing the mycobiome? I think the authors have not succeeded in communicating the idea they intended to express.
213 Whose sensitivity to what? specify
Author Response
Thank you for your help in the second round revision. Please, find below our responses to your queries and correction suggestions.
Lines 20-21 Delete the added part of the sentence ‘which is indicative of possible new findings deriving from the exploration of this particular symbiotic context.’ as this isn’t adding any information.
Done.
49 carried out -> published Delete ‘in the past’
Done.
51-54 I still don’t get what is meant by ‘depending on their traditional categorization in the guild of fungal antagonists’ and ‘'removal of the above cultural constraint’ Do you mean that Talaromyces/Penicillium species were traditionally considered as saprotrophs and therefore, even if found on insects, these were not classified as potential entomopathogens or insect mutualists?
Yes, that's what we mean.
What has changed in this regard? I don’t think this can be called ‘cultural’ but rather an improvement of our knowledge based on increasing sampling and improvement of identification methods. Please change in a way allowing unambiguous understanding.
We modified this sentence, and we hope the concept can now be understood unambiguously.
the molecular tool -> molecular tools
Done.
The sentence on lines 129-131 is impossible to understand, especially now after an addition to its end:
Moreover, the occurrence of Talaromyces resulted in the analysis of the endomycobiome of another planthopper species, Delphacodes kuscheli, even if in low relative abundance [57].
Who was observed in relative abundance and how did the observation of the fungus result in anayzing the mycobiome? I think the authors have not succeeded in communicating the idea they intended to express.
This sentence was modified in a more understandable form. The concept of relative abundance refers to the the whole mycobiome, as expressed in ref. [57].
213 Whose sensitivity to what? specify
Done.
Reviewer 2 Report
Dear authors, i believe that this in a good stage this ms. I believe that the conclusion is not clear. Please write some parts.
Author Response
We cannot modify the Conclusions section in the absence of specific remarks, also considering that Reviewer # 1 has not raised any comments on this matter.